# Kynurenines, Neuronal Excitotoxicity, and Mitochondrial Oxidative Stress: Role of the Intestinal Flora

**DOI:** 10.3390/ijms25031698

**Published:** 2024-01-30

**Authors:** Gábor Nagy-Grócz, Eleonóra Spekker, László Vécsei

**Affiliations:** 1Department of Neurology, Faculty of Medicine, Albert Szent-Györgyi Clinical Center, University of Szeged, Semmelweis u. 6, H-6725 Szeged, Hungary; nagy-grocz.gabor@szte.hu; 2Faculty of Health Sciences and Social Studies, University of Szeged, Temesvári krt. 31., H-6726 Szeged, Hungary; 3Preventive Health Sciences Research Group, Incubation Competence Centre of the Centre of Excellence for Interdisciplinary Research, Development and Innovation of the University of Szeged, H-6720 Szeged, Hungary; 4Pharmacoidea Ltd., Derkovits fasor 7., H-6726 Szeged, Hungary; spekker.eleonora@gmail.com; 5HUN-REN-SZTE Neuroscience Research Group, University of Szeged, Semmelweis u. 6, H-6725 Szeged, Hungary

**Keywords:** kynurenine pathway, intestinal flora, excitotoxicity, oxidative stress

## Abstract

The intestinal flora has been the focus of numerous investigations recently, with inquiries not just into the gastrointestinal aspects but also the pathomechanism of other diseases such as nervous system disorders and mitochondrial diseases. Mitochondrial disorders are the most common type of inheritable metabolic illness caused by mutations of mitochondrial and nuclear DNA. Despite the intensive research, its diagnosis is usually difficult, and unfortunately, treating it challenges physicians. Metabolites of the kynurenine pathway are linked to many disorders, such as depression, schizophrenia, migraine, and also diseases associated with impaired mitochondrial function. The kynurenine pathway includes many substances, for instance kynurenic acid and quinolinic acid. In this review, we would like to show a possible link between the metabolites of the kynurenine pathway and mitochondrial stress in the context of intestinal flora. Furthermore, we summarize the possible markers of and future therapeutic options for the kynurenine pathway in excitotoxicity and mitochondrial oxidative stress.

## 1. Introduction

The gut microbiota, also referred to as the intestinal flora, has become a focal point in healthcare. One of the most noteworthy advancements in recent gut microbiota research has been the revelation that the assembly of microorganisms in our digestive system has the capacity to influence various aspects of brain development and function. 

Examining the neurobiological mechanisms responsible for the degree of influence exerted by microbial organisms on host physiology, brain function, and behavior has become a top research priority. Researchers are currently exploring various pathways and potential mechanisms that may govern the interactions between intestinal flora and the brain. 

One central focus in this area of investigation is how the microbiota regulates the availability of circulating tryptophan (Trp). Trp is an essential amino acid that is responsible for the synthesis of serotonin (5-HT) and, moreover, is an initial compound of the kynurenine pathway (KP). The KP has a crucial role in many disorders, such as immunological [1], neurological [2], and mitochondrial diseases [3]. Mitochondrial disorders are chronic genetic illnesses that occur when mitochondria in cells fail to produce sufficient energy due to damage to their own DNA.

There is limited direct evidence establishing a clear and causal connection between the gut microbiota and mitochondrial disorders. However, research in both areas has been progressing, and there is growing interest in understanding potential links between them. This review focuses on the possible role of the gut microflora in the production of KP metabolites, and the role of the two systems in the pathophysiology of mitochondrial disorders. 

## 2. Connections between the Intestinal Flora and the KP

### 2.1. Intestinal Flora

Intestinal flora, also known as the gut microbiota or gut microbiome, refers to the diverse community of microorganisms that inhabit the human gastrointestinal tract, primarily the colon. These microorganisms consist of bacteria, viruses, fungi, and other microorganisms, with bacteria being the most abundant and well-studied component. The human gut is home to trillions of microorganisms, with an estimated 1000 to 1150 different species of bacteria. 

The initial colonization of microbes primarily occurs during the process of childbirth. Newborns delivered through the vaginal route are exposed to bacteria from their mother’s feces and vaginal region, while infants born via cesarean section initially encounter bacteria from the hospital environment and their mother’s skin [4]. It is important to note that despite the long-lasting belief that the prenatal environment is completely sterile, recent research has shown that before breastfeeding, the placenta, the amniotic fluid, and meconium can carry a certain amount of bacteria [5]. Studies have detected, in the infant gut, the presence of bacteria such as Enterobacteriaceae, Bifidobacterium and Bacteroides [6].

The infant gut microbiota remains highly changeable and dynamic until around the age of two when the consumption of solid foods begins [4]. Afterward, around the third year of life, as the diet becomes more diverse, the microbiota stabilizes and starts resembling the microbial compositions found in adults [5]. In adulthood, healthy people’s gut microbiota is primarily populated by four main phyla: Firmicutes, Actinobacteria, Bacteroidetes, and Verrucomicrobia [7]. The composition of the gut microbiota in healthy young adults and middle-aged individuals is characterized by a high diversity of bacterial species [8]. However, as individuals age, their gut microbiota undergoes changes, with a higher proportion of Bacteroides spp. and distinct patterns of Clostridium groups identified in the elderly compared to younger adults [9]. Bacteria in these species can also metabolize Trp into substances of the KP [10]. 

Therefore, during infancy and old age, gut microbiota composition is highly dynamic and experiences significant shifts, whereas in healthy young adults and middle-aged individuals, it tends to be more stable and diverse. Even during adulthood, the composition of the gut microbiota can undergo significant changes over the course of a year [11]. This has led to debates regarding the best way to characterize and monitor an individual’s gut microbiota composition. The concept of “enterotypes” (three core clusters of bacterial genera: Bacteroides, Prevotella, and Ruminococcus) is not universally accepted due to variations between individuals and challenges in categorizing an individual’s gut microbial composition within one specific enterotype [11]. An alternative approach suggests that the gut microbiota composition reflects a core set of functional profiles, where certain bacterial species play a more critical role in influencing health and disease [12].

This diversity plays a crucial role in maintaining a healthy gut. So gut microorganisms are an intricate microbial community recognized as a crucial influencer of neurodevelopment [13], and their involvement in the development of excitotoxicity is highly probable [14]. The potential contribution of gut microbiota to brain development can be described as follows [15]. On the one hand, Trp serves as a precursor for various biologically significant metabolites with crucial physiological roles, and it holds significant implications for the development of the central nervous system (CNS). Serotonin (5-hydroxytryptamine, 5-HT), originating from Trp via the action of Trp hydroxylase, regulates essential neurodevelopmental approaches [15]. These processes are under the influence of short-chain fatty acids (SCFAs) produced by the gut microbiome, such as acetate, propionate, and butyrate [16,17]. These substances can influence the brain by modifying the levels of neurotransmitter precursors [18] and via the vagal nerve [19]. On the other hand, neurotrophins such as brain-derived neurotrophic factor (BDNF), which are closely associated with neurodevelopment and neuroprotection, play crucial roles in promoting neuronal survival, synaptic plasticity, and cognitive function [15]. Recently, it has been also shown that the KP is changed in BDNF knock-in mice [20], so there is a connection between the substances of the KP and the BDNF pathway. Additionally, it is proposed that gut microbiota may influence the expression of N-methyl-D-aspartate (NMDA) receptors and contribute to brain development [21].

To sum up, bacteria in the gut microbiome can metabolize Trp, leading to the production of various metabolites within the gut (Figure 1). 

### 2.2. The KP and Its Receptors

Trp is an essential amino acid that is crucial in the brain as it is the precursor of 5-HT. Several bacterial strains have the capability to enhance the production of Trp in the body, potentially playing a role in maintaining regular gut motility [22]. Nonetheless, the majority (more than 90%) of Trp in mammalian cells metabolizes in the KP and not towards the 5-HT (Figure 2). Kynurenic acid (KYNA) is one of the best-known substances of the KP, which was first described by Justus von Liebig in 1853 [23]. Liebig was a German scientist, who discovered KYNA in dog urine. It is important to know that KYNA is an endogenous glutamate receptor antagonist and has neuroprotective effects. This substance is created by kynurenine aminotransferases (KATs) from L-kynurenine (L-KYN), which is constructed by the formamidase enzyme from N-formylkynurenine. N-formylkynurenine, and is formed from L-Trp by two iron-dependent enzymes: indolamine 2,3-dioxygenase 1 and 2 (IDO1 and IDO2) and tryptophan 2,3-dioxygenase (TDO). It is noteworthy that, besides KYNA, L-KYN can transform into anthranilic acid (ANA) in reaction to kynureninase (3-HAO) and to 3-hydroxykynurenine (3-HK) when reacting to kynurenine 3-monooxygenase (KMO). KMO, in eukaryotic cells, is a mitochondrial protein located in the outer membrane of mitochondria [24]. ANA can further transform into 3-hydroxyanthranilic acid (3-HAA) when exposed to 3-hydroxyanthranilic acid hydroxylase. Furthermore, 3-HK can also be converted to 3-HAA by the kynureninase enzyme. In addition to this, 3-HK can also transform into xanthurenic acid. 3-HAA can further be converted to quinolinic acid (QUIN) when exposed to 3-hydroxyanthranillic acid 3,4-dioxygenase. In the last step of the KP, QUIN is transformed into nicotinamide adenine dinucleotide (NAD^+^) in reaction to quinolinic acid phosphoribosyl transferase. NAD^+^ has a pivotal role in mitochondrial energy management and redox reactions [25]. Different from KYNA, QUIN is an endogenous glutamate receptor agonist, which when produced by microglia [26], can provoke lipid peroxidation [27] and has a relevant role in the neurodegenerative process [28]. QUIN can exert its effects, probably by stimulating NMDA receptors, yielding the overproduction of free radicals and inhibiting the respiratory chain. Research shows that the enzymes and substances of the KP have pivotal roles in the pathomechanisms of migraine [29], Parkinson’s disease [30], schizophrenia [31] and mitochondrial dysfunction [32,33,34]. 

Kynurenines exert their effects on different receptors, such as glutamate receptors in a dose dependent manner, aryl hydrocarbon (AhR) receptors, G protein-coupled receptor 35 (GPR35), and α-7 nicotinic (α7 nACh) receptors (Figure 3). Glutamate receptors are a class of neurotransmitter receptors found in the CNS of animals, hence humans. Glutamate is the most abundant excitatory neurotransmitter in the brain, and its receptors play a fundamental role in various aspects of neuronal communication and synaptic plasticity. There are several types of glutamate receptors, but they are generally categorized into two major classes. The ionotropic glutamate receptors function as ligand-gated ion channels. This implies that upon binding with glutamate, they facilitate the movement of ions, including the influx of sodium (Na^+^) and calcium (Ca^2+^) as well as the efflux of potassium (K^+^) through the cell membrane. Ionotropic receptors mediate fast synaptic transmission and include three subtypes: NMDA receptors, α-amino-3-hydroxy-5-methyl-4-isoxazolepropionic acid (AMPA) receptors, and kainate receptors. The other types of glutamate receptors are metabotropic glutamate receptors, which are G protein-coupled receptors that are indirectly linked to ion channels through intracellular signaling pathways. They modulate synaptic transmission and play a role in the regulation of neuronal excitability. KYNA is an antagonist at the strychnine-insensitive glycine-binding site of NMDA receptors at low doses [35] and can also inhibit the glutamate-binding site of the NMDA receptors at higher doses [36]. In addition to this, KYNA has an antagonistic impact on kainate- and AMPA receptors [35]. The effect of KYNA on AMPA receptors is also concentration dependent, which means that KYNA can stimulate the receptors at nanomolar and micromolar concentrations but in different circumstances, namely between micromolar and millimolar concentrations, can inhibit the AMPA receptors [37,38].

AhR receptors are a family of proteins found in many species, including humans. AhR is a ligand-activated transcription factor that can bind to various aromatic hydrocarbons, including environmental pollutants like dioxins and polycyclic aromatic hydrocarbons (PAHs). When AhR binds to its ligands, it undergoes a conformational change and translocates to the cell nucleus, where it regulates the expression of various genes. The AhR receptor activation promotes equilibrium in host-microbe interactions, with indole derived from microbial Trp serving as a crucial ligand for this transcription factor [39]. Additionally, the AhR receptor is involved in the regulation of IDO and TDO expression [40,41]. Elevated AhR receptor activation has been noted in individuals with post-acute sequelae of SARS-CoV-2 infection, correlating with heightened IDO2 activity [42]. Additionally, a decline in mitochondrial function is evident in these patients. Experimental findings indicate that AhR receptor antagonists can diminish IDO2 activation [42]. Consequently, it can be inferred that modifications in the KP might impact mitochondrial function through AhR receptors.

GPR35 is a member of the G protein-coupled receptor family. The exact function of GPR35 is not fully understood, and it appears to have diverse roles in different tissues. Research has implicated GPR35 in immune responses, inflammatory processes, and gastrointestinal functions.

The α7 nACh receptor is a type of nicotinic acetylcholine receptor that plays a crucial role in neurotransmission within the CNS. Nicotinic receptors are ionotropic receptors that respond to the neurotransmitter acetylcholine, and they are named after nicotine, which also binds to and activates these receptors. Doubts regarding the effects of KYNA on α7 nACh receptors have been raised by Stone [43].

### 2.3. Role of the Intestinal Flora in the KP Metabolism

The relationship between diet, the KP, and the gut microbiome is a complex and emerging area of research in the fields of nutrition, metabolism, and microbiology. Various diets can significantly influence the composition and function of the gut microbiome [44]. For example, diets rich in fiber from fruits, vegetables, and whole grains can promote the growth of beneficial bacteria in the gut [45]. On the other hand, high-fat and high-sugar diets can alter the microbiome composition and may contribute to the development of metabolic disorders [46].

Research has shown that Trp, and thus the KP, has a unique role in the dual communication between microorganisms of the gut and the various organs of the host [47]. Bacteria in the gut can influence the activity of IDO1 in the gut, and some of these bacteria have enzymes very similar to those of the KP and thus contribute to the production of the metabolites of the KP [47]. It is widely recognized that the bidirectional communication between the microbiota and the immune system plays a crucial role in shaping the host’s intestinal immune response [48]. Consequently, mice that lack gut microbiota display a deficiency in their innate immune system. When compared to conventional mice, germ-free mice exhibit decreased degradation of Trp via the KP, leading to higher levels of available Trp and lower levels of KYNA in their bloodstream [49,50]. Additionally, the levels of circulating Trp and KYNA return to normal after microbial colonization in mice immediately after weaning [50].

In a recent study, Sun and his colleagues found that long-term high-fat diets disrupted the metabolism of Trp in the bloodstream in mice [51]. This disruption was characterized by a reduction in Trp levels and an increase in the activity of the KP. Notably, this aberrant Trp metabolism strongly correlated with the proliferation of the Proteobacteria phylum in the colon, so deviations in Trp metabolism can induce changes in the KP metabolism [51] as well. In this interesting research, the authors could abrogate the long-term high-fat diet shift with antibiotic treatment, but changes induced by the long-term high-fat diet could be transferred to mice with a standard diet through fecal transplantation, thus revealing the role of gut microbiota in this process. In another experimental setting, it was shown that the administration of Bacillus infantis to rats elevates Trp and KYNA levels but decreases the level of 5-Hydroxyindoleacetic acid [52], which is the main metabolite of 5-HT [53]. These data obviously prove that the gut microbiome influences the KP. 

Alterations in the KP have been documented in several diseases linked to a disrupted microbiome. Patients diagnosed with irritable bowel syndrome (IBS) have reported heightened KP metabolism and altered microbiome [54,55,56], providing evidence of a connection between these two systems. The roles of altered microbiome and oxidative stress have also been described in patients with Crohn’s disease [57,58]. In a recent study, it has been shown that Trp has a protective impact on Crohn’s disease and IBS [59], by strengthening the role of the KP against mitochondrial stress and supporting a normal microbiome. 

However, it is important to note that the interplay between diet, kynurenine, and the gut microbiome is still an active area of research and that the mechanisms involved are not fully understood. It is also important to consider individual variations in microbiome composition and responses to diet.

## 3. Mitochondrial Disorders, and Oxidative Stress: Possible Role of the KP

### 3.1. Mitochondria

Mitochondria are the provider and warehouse of energy in eukaryotic organ-isms; they have a cylinder shape, and almost every cell (except red blood cells) contains thousands of them. The origin of mitochondria is described by the endosymbiotic theory, which states that in the distant past a proteobacterium was swallowed up via endocytosis, affording the host with the skill to produce energy in the form of adenosine triphosphate (ATP). This double-membrane cell-forming cell has no nucleus, but it has its own genetic material called (mtDNA). The role of mitochondria is essential in many cellular processes, e.g., calcium homeostasis [60], apoptosis [61], free radical formation, and, of course, the production of ATP via oxidative phosphorylation. In this process, electrons are transported to molecular oxygen via the mitochondrial respiratory chain, namely complex I to V, which develops a proton gradient through the mitochondrial membrane. Thereafter, the complex V (or ATP synthase) can generate ATP. Malfunction of the oxidative phosphorylation process can increase production of highly reactive free radicals (redox homeostasis), yielding oxidative stress, lipid peroxidation, DNA damage, and cell death.

### 3.2. Mitochondrial Disorders and Oxidative Stress

Mitochondrial disorders are chronic, uncurable conditions, the diagnosis of which is very difficult. These disorders can affect various organs and systems in the body, and their severity can vary widely. They have many, non-specific symptoms, from muscle weakness to seizures (Figure 4). 

The causes underlying mitochondrial disorders include gene mutations as well as environmental factors. These mutations can affect both nuclear and mitochondrial DNA, with more than 350 genes currently identified as bases for mitochondrial diseases [62]. We can divide mitochondrial disorders into primary and secondary mitochondrial diseases. Primary mitochondrial conditions (Figure 5) are genetically determined and identified by mutations in the nuclear or mtDNA.

In addition to this, secondary mitochondrial disorders (Figure 5) are triggered by mutations or several environmental factors, such as alcohol [63], smoking [64], carbon monoxide [65], and antiretroviral [66] or aminoglycoside [67] therapy.

What increases the difficulty of diagnosis is that we have no specific laboratory or diagnostic tests to prove the diagnosis of mitochondrial disorders. The age of onset, affected organs, and severity of symptoms can vary significantly among individuals. Symptoms can involve the nervous system, muscles, heart, liver, kidneys, and other organs [68]. The first patient with mitochondrial disorder was described by Luft in 1959 [69]. He examined a 30-year-old woman with exudation and raised fluid intake, full-body weakness, and inability to gain weight despite a huge caloric intake. The symptoms of the patient began at the age of 7. Laboratory tests showed normal thyroid function and that the basal metabolic rate was extremely high. Muscle biopsy showed high ATPase activity, elevated level of cytochrome oxidase, and low levels of coenzyme Q in the mitochondria, which strongly pointed towards intensified mitochondrial synthesis.

The diagnosability of these disorders has improved recently, with the most trustworthy method being genetic testing. The first step is usually the examination of mtDNA, and then analyses of nuclear DNA for genes affected in mitochondrial diseases. It is possible to completely examine the nuclear DNA with whole genome sequencing. In addition to this, clinicians can also use additional tests, such as a muscle biopsy, biochemical tests on blood, urine, cerebrospinal fluid, and MRI [70]. Despite the intensive research, we have no completely reliable testing methods [71] that can be used in all patients suspected of mitochondrial disorder.

At present, an effective and targeted treatment for the majority of patients with mitochondrial disease is still unavailable. Various approaches, primarily involving nutritional supplements like creatine, co-enzyme Q10, carnitine, and vitamin combinations, have been extensively utilized, relying on individual case reports [72]. Treatment also focuses on managing symptoms and supportive care, including physical therapy and occupational therapy [73].

In conclusion, mitochondrial disorders represent a complex group of genetic diseases with diverse clinical presentations. Ongoing research is essential for advancing our understanding and developing potential therapeutic interventions for these challenging conditions.

Oxidative stress is a cellular condition that occurs when there is an imbalance between the production of reactive oxygen species (ROS) and the body’s ability to eliminate them or neutralize their harmful effects. ROS, including free radicals like superoxide and hydrogen peroxide, are natural products of metabolic processes, including those in the mitochondria. Under normal conditions, the body has defense mechanisms to contain ROS and prevent cellular damage. However, when this balance is disrupted, oxidative stress can occur, leading to damage to lipids, proteins, and DNA within cells. Oxidative stress, if not managed effectively by the cell’s antioxidant defenses, can lead to cellular damage, including damage to mitochondrial components such as mtDNA. This can further exacerbate mitochondrial dysfunction and create a vicious cycle of increased ROS production and cellular damage; thus, mitochondria patients can suffer continuous ROS harm.

Mitochondrial stress and intestinal flora exhibit a complex relationship that influences overall health and homeostasis in the body. In response to elevated glucose stress, the irregular structure and/or function of mitochondria can trigger the creation of a significant quantity of ROS [74] by driving mitochondria towards heightened oxygen consumption and increased redox potential [75]. Additionally, hyperglycemia alters oxygen transport, favoring the respiratory chain complex II [76]. The process of converting ADP to ATP reduces membrane potential, but hyperglycemia promotes ADP regeneration and consumption rates. This leads to a decline in ATP formation, a persistent rise in membrane potential, and an amplification of ROS production [77]. These ROS can activate ryanodine receptor 2 and suppress the activity of sarcoplasmic reticulum calcium transport ATPase [78], resulting in elevated apoptosis levels and reduced mitophagy levels [79], which contribute to mitochondrial damage [80]. Mitochondrial dysfunction can compromise the integrity of the gut barrier [81]. The gut barrier prevents the entry of harmful substances into the bloodstream and plays a crucial role in immune regulation. Mitochondrial stress-induced damage may lead to increased intestinal permeability, allowing the translocation of bacterial components and toxins [82].

To sum up, there is no direct link between mitochondrial stress and intestinal flora; however, some studies suggest that the gut microbiota can influence host mitochondrial function indirectly through several mechanisms as previously summarized. On the one hand, gut bacteria can produce SCFAs through the fermentation of dietary fiber [83]. SCFAs have been shown to positively impact mitochondrial function and overall cellular health [84]. On the other hand, dysbiosis, an imbalance in the gut microbiota, can lead to inflammation [85]. Chronic inflammation may affect mitochondrial function and contribute to mitochondrial stress. In addition, the gut microbiota produces various metabolites that can enter the bloodstream and potentially affect distant organs, including mitochondria [81]. Finally, the gut microbiota plays a role in regulating the immune system. Immune responses, particularly inflammation, can impact mitochondrial function [86].

### 3.3. The Role of the Kynurenine Pathway in the Pathophysiology of Mitochondrial Diseases

The last step of the KP is the conversion of QUIN to NAD^+^, which plays a regulatory role in various cellular functions, including mitochondrial function, DNA repair, and the response to hypoxia by adjusting the cell’s energy state [87,88,89]. Additionally, intermediate activity within the KP can also affect NAD^+^ conversion and mitochondrial action by influencing levels of ROS [25]. The prevailing understanding suggests that the primary role of the KP is the generation of NAD^+^, a molecule closely associated with the control of critical oxidative stress processes. In addition, elevated KMO expression results in the formation of intermediary substances that trigger glutamate receptors, culminating in oxidative stress [90]. Moreover, within the peripheral nervous system, 3-HK is transported across the blood-brain barrier (BBB), consequently enhancing its presence and availability in the brain [91]. Both in vivo and in vitro investigations have revealed that the neuronal cell death induced by 3-HK is alleviated by antioxidants like glutathione, catalase, and deferoxamine [92]. These findings illustrate that the detrimental impacts of 3-HK are manifested by imposing oxidative stress.

KYNA and QUIN constitute a vital pair responsible for upholding equilibrium in oxidative stress levels [93]. QUIN’s neurotoxicity arises from heightened glutamate signaling, lipid peroxidation, and its role as an activator of NMDA receptors [94], so increased levels of QUIN lead to NMDA receptor activation. Findings from animal research have validated that QUIN triggers inflammation and oxidative stress in rats, potentially resulting in neurodegeneration within the striatum [95]. In contrast, KYNA counteracts the neurotoxic impact of QUIN and serves as a physical shield for neurons [96]. Consequently, maintaining the balance of metabolites within the KP is imperative for preserving normal mitochondrial function (Figure 6).

Some of the KP metabolites can influence mitochondrial dynamics, particularly the fusion and fission cycles. These processes can regulate the function and morphology of the mitochondria because the damaged mitochondria can fuse with normal neighboring mitochondria, resulting in its reconstruction. The process of fission means the transport of seriously damaged mitochondria to lysosomes for breakdown, a process also called mitophagy. Mitophagy represents a specialized variant of autophagy designed to discern and eliminate impaired or unneeded mitochondria that features essential components and key proteins shared with the broader autophagic process. The unique mechanism responsible for enveloping mitochondria within phagophores is established through separate mechanisms [97]. Forms of mitophagy include PINK1-PRKN axis-dependent mitophagy, receptor-mediated mitophagy, FUNDC1-mediated mitophagy, BNIP3- and BNIP3L-mediated mitophagy, BCL2L13-mediated mitophagy, FKBP8-mediated mitophagy, and PHB2-mediated mitophagy, as discussed in Zhang’s review article [97]. These processes representing the dynamics of mitochondria have a crucial role in cellular homeostasis [98]. Several metabolic, cardiovascular problems, and neurodegenerative diseases can be traced back to the malfunctioning of these dynamics (Figure 7). 

Mitochondrial fission is modulated by the fission 1 protein and dynamin related protein (Drp1). It is important to note that the overexpression of KMO can regulate the phosphorylation of Drp1, elevating mitochondrial fission [34]; so KMO can influence the dynamics and homeostasis of mitochondria. In addition to this, knockdown of the KMO gene causes a distortion of shape in mitochondria [99] and increases the number of mitochondria with reduced respiratory activity. Taken together, KMO regulates mitochondrial dynamics, probably via interaction with Drp1 [34].

On the other hand, metabolites of the KP can influence the level of NAD^+^, which has a role in mitochondrial function, dynamics, and the production of free radicals [25]. Supplemental Trp can elevate lifespan in mice [100] and in Caenorhabditis elegans [101], probably by increasing the level of NAD^+^ [102].

Leigh syndrome, the most common mitochondrial disease in childhood, is a neurodegenerative disorder caused by over 75 known genetic mutations [103]. In this paper, the authors provide a comprehensive overview of the genetic, biochemical, clinical, metabolic and neuroradiological heterogeneity of Leigh syndrome. A recent study revealed that individuals with French Canadian variants of Leigh syndrome exhibited decreased levels of L-KYN and 3-HAA in their blood [104]. Furthermore, in these patients, there was an elevation in indoxyl sulfate levels [104], indicating a potential shift in Trp metabolism toward the indole pathway. It is worth mentioning that Trp can undergo metabolism not only to kynurenine or 5-HT but also to indoxyl sulfate. Trp can be transformed into indole by tryptophanase, and indole can further metabolize to indoxyl through cytochrome P450 2E1. Subsequently, sulfotransferase enzymes can convert indole to indoxyl sulfate.

Leber hereditary optic neuropathy is characterized by the degeneration of retinal ganglion cells, leading to vision loss. Patients with this condition typically carry mutations in genes that encode complex I within the electron transport chain. Studies of Leber hereditary optic neuropathy patients have identified reduced levels of Trp and glutamate [105], indicating potential involvement of the KP in the disease’s pathomechanism. Trp is the precursor of the KP, as mentioned above, and KYNA is a glutamate receptor antagonist. Myoclonic epilepsy with ragged red fibers is a primary mitochondrial disorder. This is an extremely rare illness, and can affect different parts of the body, especially the nervous system and the muscles. One of the main symptoms of the disorder is myoclonus epilepsy, and we know that 5-HT and the KP pathways change in patients with Unverricht–Lundborg disease, which is a type of progressive myoclonus epilepsy, and in cystatin B (CSTB)-deficient mice [106].

A recent study has shown that the metabolites of the KP can be used as biomarkers of several mitochondrial diseases. Buzkova and her colleagues found a decrease of KYNA and niacinamid balance in the blood of infantile-onset spinocerebellar ataxia patients, which suggests altered NAD^+^ production and elevated NAD^+^ demand [107]. This is a severe, progressive neurodegenerative condition that appears usually at the age of 1. The symptoms of the disease are ataxia, muscle hypotonia, ophthalmoplegia, deafness, and epileptic seizures. In addition to this, researchers also found an increase in KYNA and 3-HK levels in patients with progressive external ophthalmoplegia and inclusion body myositis [107]. Progressive external ophthalmoplegia attacks and causes weakness in eye muscles. This condition usually has several accompanying symptoms, such as deafness, neuropathy, ataxia, parkinsonism, and depression. 

The other illness that correlates with mitochondrial dysfunction is chronic kidney disease (CKD), which is a general name for functional or structural malfunction of kidneys. Research has shown that CKD is also connected to the KP. For instance, increased L-KYN, KYNA, ANA levels were found in the plasma of patients with CKD [108]. 

Sporadic inclusion body myositis is a complex disease that has inflammatory, degenerative, and mitochondrial aspects [109]. The main manifestation of the disorder is debility of limbs. 

Some neurodegenerative disorders are also linked to mitochondrial abnormalities, oxidative stress, and alterations of the KP metabolism [110,111]. Reduced levels of KYNA have been found in the plasma of patients with Alzheimer’s disease [112], and the same changes have been detected in patients with Parkinson’s and Huntington’s disorders [113]. This observation holds significant importance as reduced KYNA levels may trigger insufficient anti-inflammatory reaction, leading to heightened tissue damage and excessive cell proliferation in inflammatory conditions [114,115], which can be seen in neurodegenerative disorders. In addition to this, elevated levels of QUIN in the cerebrospinal fluid have been observed in patients with AIDS-associated dementia [116,117]. This finding confirms the existence of heightened mitochondrial damage linked to alterations in KYNA-QUIN levels. 

In conjunction with this information, it is noteworthy that cannabinoid receptor agonists can hinder QUIN-induced mitochondrial dysfunction, lipid peroxidation, and the formation of ROS [118]. It is also important to highlight that KYNA, in an indirect and region-specific manner within the brain, enhances the presence of functional cannabinoid receptor 1 without altering the overall binding and activity of the receptor [119].

Data from animal models have pointed out that KYNA and its analogues can improve mitochondrial dysfunction in the liver, produced by sepsis, probably via the reduction of mitochondrial dysfunction of the CNS [120]. The synthetic compound mitochonic acid-5 (MA-5) is a candidate drug for the treatment of sporadic inclusion body myositis. This substance can ameliorate cell survival, elevate the level of ATP, and repair the dynamics of mitochondria in samples from patients with this disease [121]. MA-5 is also effective in the survival of fibroblasts sampled from patients with Leigh syndrome, MELAS (myopathy encephalopathy lactic acidosis and stroke-like episodes), Leber’s hereditary optic neuropathy, and Kearns-Sayre syndrome [122]. It is worthy of note that MA-5 is a derivative of indole-3-acetic acid [121], which is a plant hormone. The precursor of indole-3-acetic acid is Trp, which is also a precursor of the KP, so MA-5 and Trp have a common formation pathway. Another recent animal study has shown that kynurenines influence mitochondria death and free radical accumulation in mice after intracerebral hemorrhage through aryl hydrocarbon receptors [123] (Figure 8).

Regardless, it is important to recognize that KYNA has restricted passage through the BBB. Potential therapeutic approaches could involve aiding penetration of the BBB and impeding or delaying the release of KYNA from the CNS. Elevating KYNA levels can be achieved by administering L-KYN. However, heightened KYNA levels come with increased elimination. Various methods can inhibit this process, and one potential agent with promise is probenecid. Animal studies have demonstrated its success in maintaining elevated KYNA levels [124,125,126]. Different chemical modifications can ease the passage of KYNA through the BBB. The resulting compounds, known as analogues, exhibit a heightened affinity for crossing the BBB. Examples of such analogues include 7-chloro-KYNA and 4-chloro-L-KYN or AV-101. Although 7-chloro-KYNA does not exhibit significantly improved BBB crossing compared to KYNA, 4-chloro-L-KYN serves as a BBB-crossing precursor for 7-chloro-KYNA. Additionally, 4-chloro-L-KYN has the ability to inhibit the synthesis of QUIN [127], suggesting its potential as a viable target for alleviating the adverse effects QUIN imposes on mitochondria. It is crucial to emphasize that 4-chloro-L-KYN can be generated through bacterial synthesis, specifically by the marine bacterium Saccharomonospora sp. CNQ-490 [128]. This strain has the capability to convert Trp into 4-chloro-L-KYN within three steps [128]. This development could potentially pave the way for utilizing microbiome-based therapy in addressing various neurodegenerative diseases.

## 4. Conclusions

In summary, the gut microbiota not only affect the nervous system but also play a significant role in the functioning of the body. The connection between the gastrointestinal system and the kynurenine pathway may yield previously unseen results in the treatment of various diseases; influencing the gut flora could provide a new therapeutic target for managing different conditions.

## Figures and Tables

**Figure 1 ijms-25-01698-f001:**
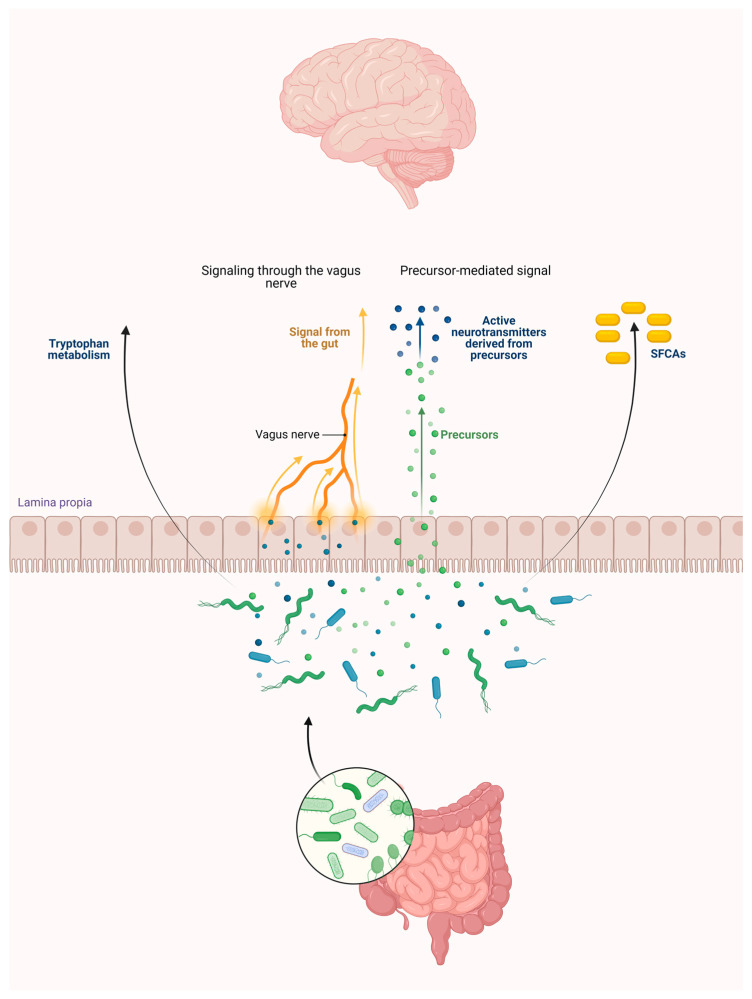
The gut microbiota influences the brain through tryptophan, short-chain fatty acids via the vagus nerve, and the precursor of neurotransmitters. Abbreviation: SCFAs—short-chain fatty acids.

**Figure 2 ijms-25-01698-f002:**
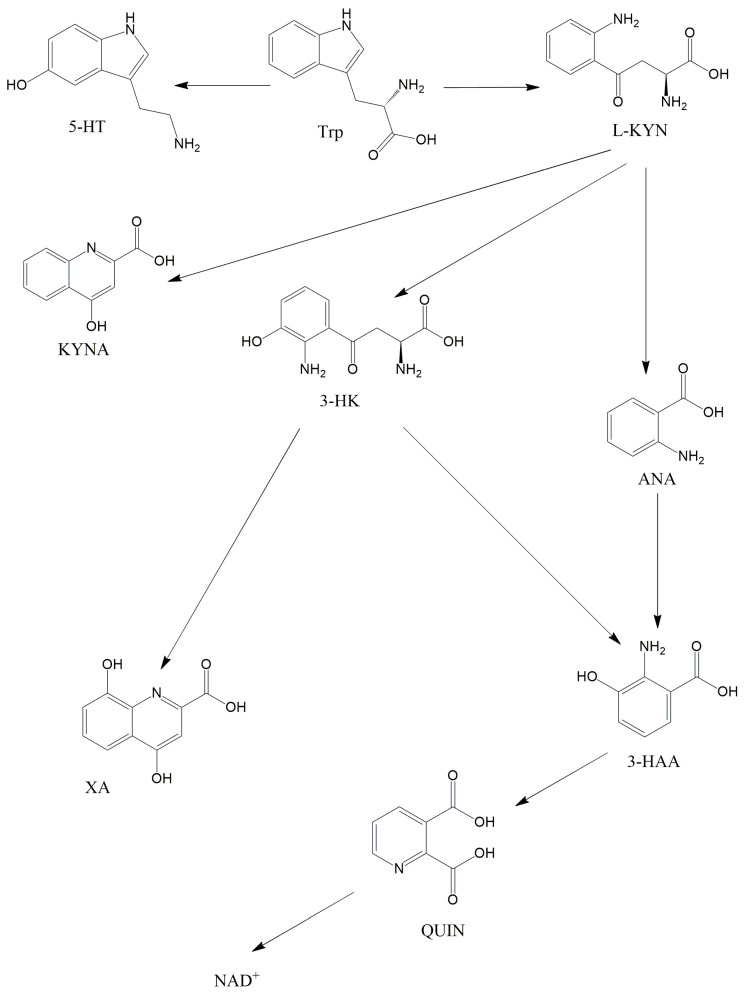
The kynurenine pathway. This figure illustrates the main metabolites of the KP. Abbreviations: 3-HAA—3-hydroxyanthranilic acid, 3-HK—3-hydroxykynurenine, 5-HT—serotonin, ANA—anthranilic acid, KYNA—kynurenic acid, L-KYN—L-kynurenine, NAD^+^—nicotinamide adenine dinucleotide, QUIN—quinolinic acid, Trp—tryptophan, XA—xanthurenic acid.

**Figure 3 ijms-25-01698-f003:**
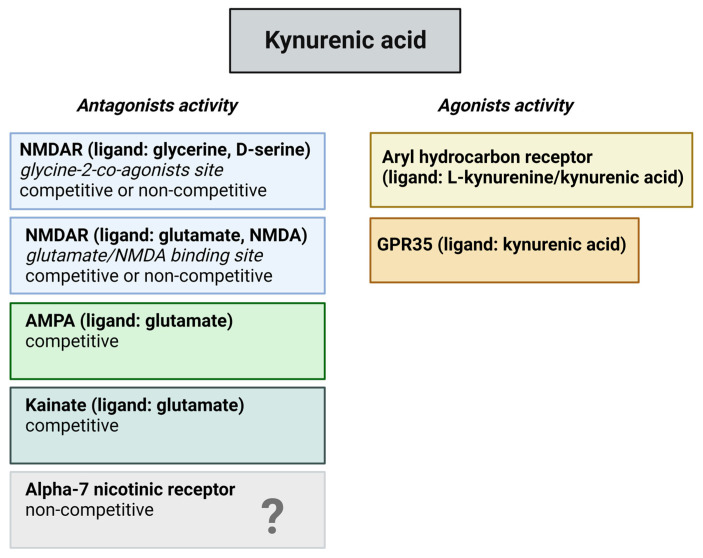
Potential binding sites for kynurenic acid. The figure provides an overview of potential receptors for kynurenic acid and their corresponding actions. Recent inquiries have raised uncertainties regarding its impact on the alpha-7 nicotinic receptor. Abbreviations: NMDAR—N-methyl-D-aspartate receptors, AMPA—α-amino-3-hydroxy-5-methyl-4-isoxazolepropionic acid receptors, GPR35—G protein-coupled receptor 35.

**Figure 4 ijms-25-01698-f004:**
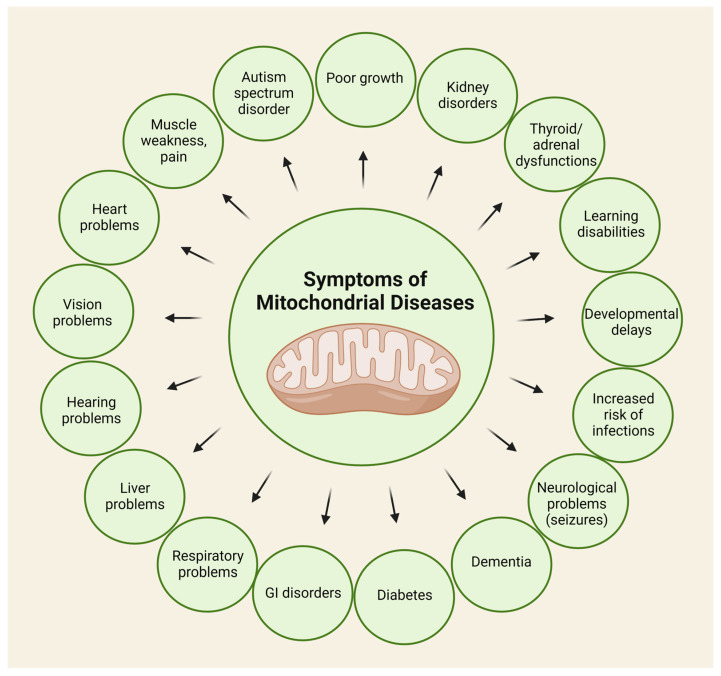
Potential symptoms of mitochondrial disorders. This figure shows that almost all organs can be affected.

**Figure 5 ijms-25-01698-f005:**
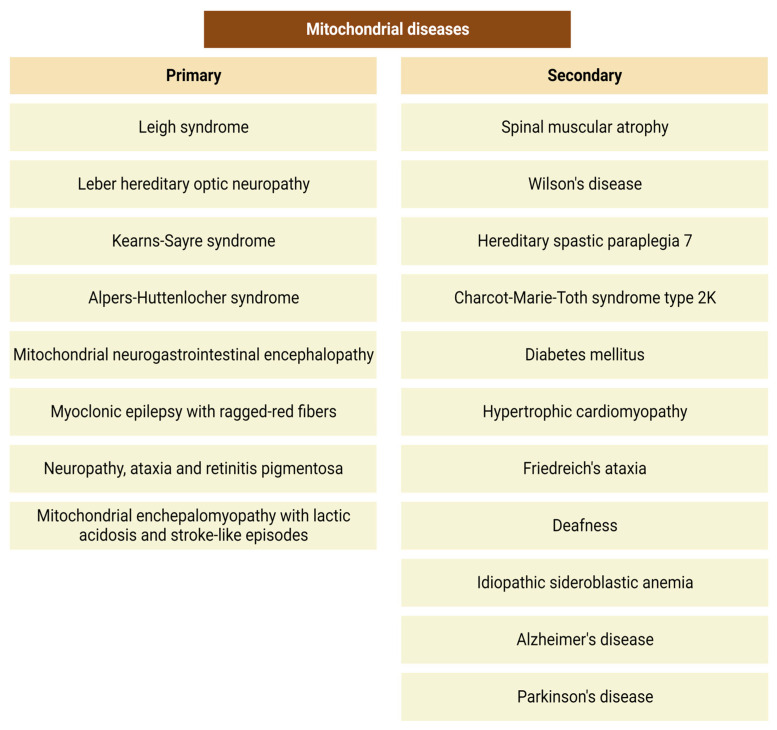
Most common primary and secondary mitochondrial disorders.

**Figure 6 ijms-25-01698-f006:**
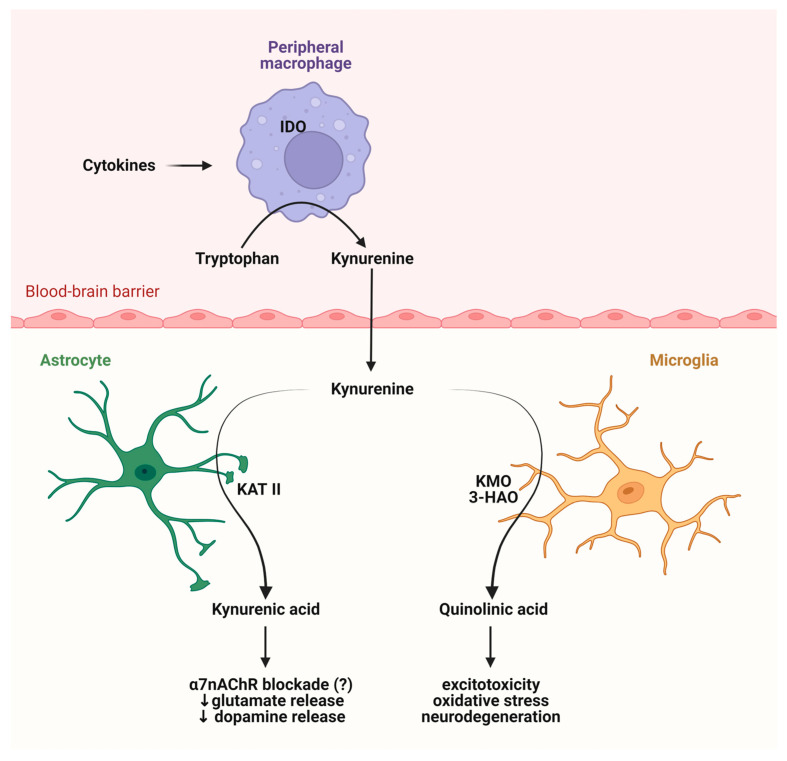
The role of the KP metabolites in the CNS. The KP plays a major role in the immune system, and immune cells can influence the production of the KP metabolites. Abbreviations: IDO—indolamine 2,3-dioxygenase, KAT II—kynurenine aminotransferase II, KMO—kynurenine 3-monooxygenase, 3-HAO—kynureninase, α7 nAChR—alpha7 nicotinic acetylcholine receptor.

**Figure 7 ijms-25-01698-f007:**
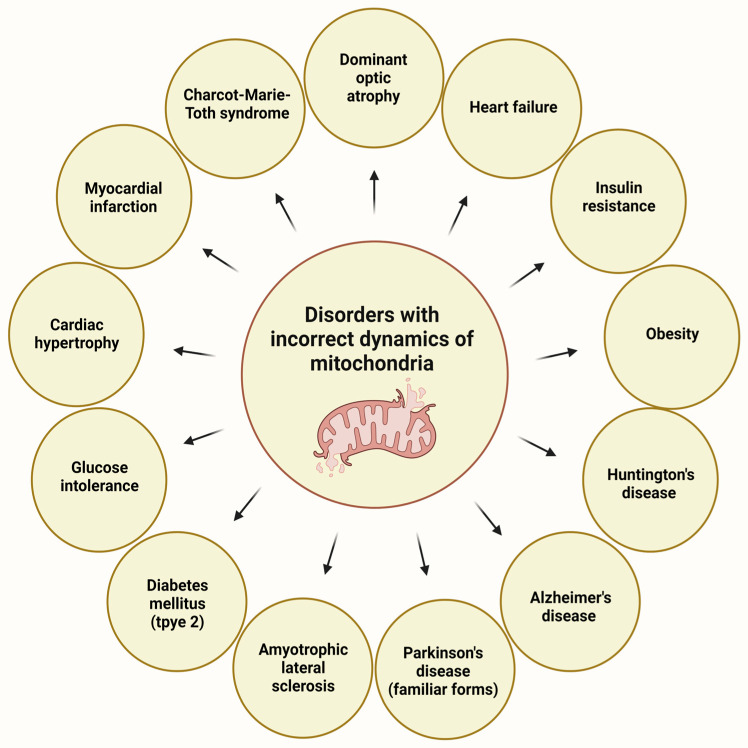
Disorders with impaired mitochondrial dynamics [3].

**Figure 8 ijms-25-01698-f008:**
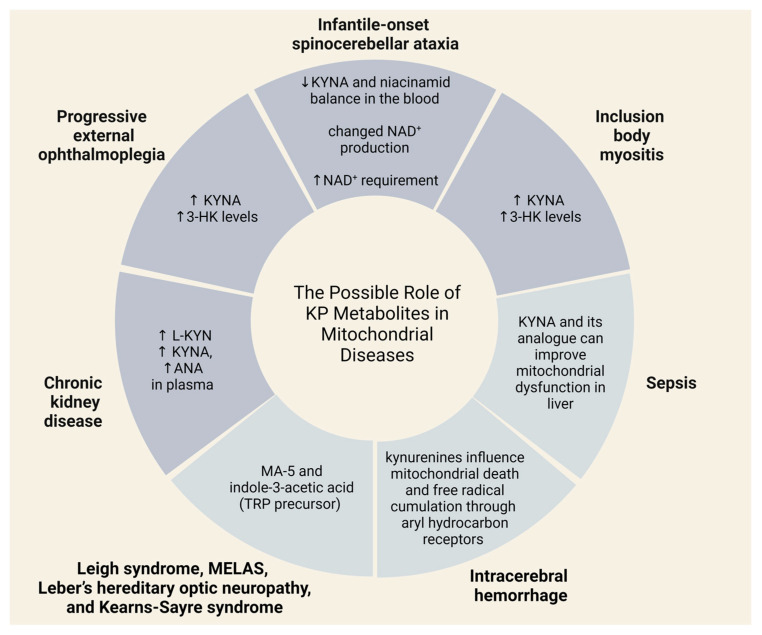
Possible roles of KP metabolites in mitochondrial diseases. Abbreviations: KYNA—kynurenic acid, NAD^+^—nicotinamide adenine dinucleotide, 3-HK—3-hydroxykynurenine, MA-5—mitochonic acid-5, TRP—tryptophan, L-KYN—l-kynurenine, ANA—anthranilic acid.

## Data Availability

Not applicable.

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
