# Peer review of "Kynurenines, Neuronal Excitotoxicity, and Mitochondrial Oxidative Stress: Role of the Intestinal Flora"

_ijms, 2024, doi:10.3390/ijms25031698_

Round 1

Reviewer 1 Report

Comments and Suggestions for Authors

Highly recommend adding more the  novel link between mitochondrial stress in context of intestinal flora, as this manuscript overlaps by recent review articles in this issue!

Additional comments:

Authors  summarized  the possible markers and future therapeutic options of the kynurenine pathway in excitotoxicity and mitochondrial oxidative stress. The topic is relevant but nothing so special or new words in this field.

The subject area compared with other published material is overlapped by many other review articles in this discipline and not have any scientific new words!

The methodology should be improved by mentioning  more details , and establishing the overall trustworthiness of qualitative other and more  research results. Highly recommend adding more the  novel link between mitochondrial stress in context of intestinal flora, as this manuscript overlaps by recent review articles in this issue. 

The references are appropriate

Author Response

Highly recommend adding more the  novel link between mitochondrial stress in context of intestinal flora, as this manuscript overlaps by recent review articles in this issue!

We extend our sincere gratitude for the thorough and professional review aimed at enhancing the quality of our manuscript. We appreciate the insightful suggestions and hope that the responses and modifications made meet with approval. Thank you once more for your valuable input.

Additional comments:

Authors summarized the possible markers and future therapeutic options of the kynurenine pathway in excitotoxicity and mitochondrial oxidative stress. The topic is relevant but nothing so special or new words in this field.

The subject area compared with other published material is overlapped by many other review articles in this discipline and not have any scientific new words!

The methodology should be improved by mentioning  more details, and establishing the overall trustworthiness of qualitative other and more  research results. Highly recommend adding more the novel link between mitochondrial stress in context of intestinal flora, as this manuscript overlaps by recent review articles in this issue.

The references are appropriate

Thank you for the reviewer's suggestion. We have supplemented the manuscript according to the request. Thank you once again for your help!

Reviewer 2 Report

Comments and Suggestions for Authors

Submitted review is a well written and informative summary on the putative role of microbiota and tryptophan metabolites in the pathogenesis of diseases linked with mitochondrial dysfunction.

I have some minor comments:

lines 93-94 - please, develop or rewrite the sentence: "The potential functions of gut microbiota in brain development can be outlined as follows."

lines 99-101 the content of the sentence does not support the involvement of microbiota in Trp metabolism, please remove  or alter

"On the other hand, neurotrophins such as brain-derived neurotrophic factor (BDNF), which are closely associated with neurodevelopment and neuroprotection, play crucial roles in promoting neuronal survival, synaptic plasticity, and cognitive function [15]."

Line 152-154, please, indicate clearly whether ion movements across the channel are inward or outward (influx or efflux)

 "The ionotropic glutamate receptors are ligand-gated ion channels, meaning
that when glutamate binds to them, they allow for the flow of ions such as sodium (Na+), potassium (K+), and calcium (Ca2+) across the cell membrane."

line 166 please, correct "micromo-lar" into "micromolar"

line 168 please, correct "Ah" into "AhR"

Comments on the Quality of English Language

lines 93-94 - please, develop or rewrite the sentence: "The potential functions of gut microbiota in brain development can be outlined as follows."

line 160 - please, change "KYNA is antagonist" into "KYNA is an antagonist"

line 247 - please, change the sentence "The causative agents of the disorders are mutations in genes and may also be caused by nongenetic impact, such as environmental factors" into: "The causes underlying mitochondrial disorders include gene mutations as well as environmental factors".

Author Response

Submitted review is a well written and informative summary on the putative role of microbiota and tryptophan metabolites in the pathogenesis of diseases linked with mitochondrial dysfunction.

We greatly appreciate your precious time and all your assistance and support, and we would be delighted to receive any additional feedback that can contribute to the improvement of our work.

I have some minor comments:

lines 93-94 - please, develop or rewrite the sentence: "The potential functions of gut microbiota in brain development can be outlined as follows."

lines 99-101 the content of the sentence does not support the involvement of microbiota in Trp metabolism, please remove  or alter

"On the other hand, neurotrophins such as brain-derived neurotrophic factor (BDNF), which are closely associated with neurodevelopment and neuroprotection, play crucial roles in promoting neuronal survival, synaptic plasticity, and cognitive function [15]."

Line 152-154, please, indicate clearly whether ion movements across the channel are inward or outward (influx or efflux)

 "The ionotropic glutamate receptors are ligand-gated ion channels, meaning

that when glutamate binds to them, they allow for the flow of ions such as sodium (Na+), potassium (K+), and calcium (Ca2+) across the cell membrane."

line 166 please, correct "micromo-lar" into "micromolar"

line 168 please, correct "Ah" into "AhR"

Comments on the Quality of English Language

lines 93-94 - please, develop or rewrite the sentence: "The potential functions of gut microbiota in brain development can be outlined as follows."

line 160 - please, change "KYNA is antagonist" into "KYNA is an antagonist"

line 247 - please, change the sentence "The causative agents of the disorders are mutations in genes and may also be caused by nongenetic impact, such as environmental factors" into: "The causes underlying mitochondrial disorders include gene mutations as well as environmental factors".

We appreciate the reviewer for pointing this out; we have modified the text with the suggestions.

Reviewer 3 Report

Comments and Suggestions for Authors

This is a review article about the influence of gut microbiome on the Kynurenine pathway (KP) of tryptophan (Trp) metabolism other than as a precursor to serotonin. The authors have done a nice job of showing how the KP generates molecules that can affect multiple neurotransmitter systems and thus affect brain function. They provide no information as to the blood-brain barrier penetration of any molecules generated by the KP. The section on mitochondrial diseases is weak. This arises from the vast expanse of mitochondrial diseases (which the authors allude to indirectly), such that any attempt to discuss mitochondrial diseases requires much greater space and referencing than the authors provide. Theirs, however, is a noble effort, and I suggest that they limit their presentation to a few specific mitochondrial conditions as examples of how the KP can affect mitochondrial function. 

Their referencing is also spotty in places. For instance, they provide no references for the statement that small chain fatty acids (SCFA) communicate with the brain through the vagus nerve. There are also several other places where they make broad statements without providing any references (even to other review articles). And, in fairness, in other places they make broad statements that DO refer to other review articles. Overall, especially for a review article, they need to tighten up their referencing.

They also don't do an adequate (by that I mean mainly consistent) job in linking the microbiome metabolism of Trp to brain function. They describe what is possible without reviewing what may actually occur (or what has been shown to occur). They do refer to a few situations where alteration of microbiomes does affect KP metabolite levels, but surely there is much more they could present.

In conclusion, this is a good review article for what it provides, but there are deficiencies in areas where it doesn't provide adequate information. As the authors are aware, the area of microbiome physiology is rapidly developing, and good review articles along the way can direct research efforts. 

As an aside, I particularly enjoyed their Figures, mainly for their simplicity and effective communication of ideas as opposed to details.

Comments on the Quality of English Language

Overall the English is good, but there are several places where subject-verb inconsistencies are present. These can be resolved by careful editing.

Author Response

This is a review article about the influence of gut microbiome on the Kynurenine pathway (KP) of tryptophan (Trp) metabolism other than as a precursor to serotonin. The authors have done a nice job of showing how the KP generates molecules that can affect multiple neurotransmitter systems and thus affect brain function. They provide no information as to the blood-brain barrier penetration of any molecules generated by the KP. The section on mitochondrial diseases is weak. This arises from the vast expanse of mitochondrial diseases (which the authors allude to indirectly), such that any attempt to discuss mitochondrial diseases requires much greater space and referencing than the authors provide. Theirs, however, is a noble effort, and I suggest that they limit their presentation to a few specific mitochondrial conditions as examples of how the KP can affect mitochondrial function.

Thanks to the reviewer for bringing this to our attention. We supplemented the manuscript with some additional information about the passage of kynurenins through the blood-brain barrier, as the reviewer request. In addition to this, we have supplemented the manuscript with more information about mitochondrial diseases. Although, it should be noted that these are shown in chapter: 3.4. The possible role of kynurenines in diagnosing mitochondrial diseases.

Their referencing is also spotty in places. For instance, they provide no references for the statement that small chain fatty acids (SCFA) communicate with the brain through the vagus nerve. There are also several other places where they make broad statements without providing any references (even to other review articles). And, in fairness, in other places they make broad statements that DO refer to other review articles. Overall, especially for a review article, they need to tighten up their referencing.

Thank you for the reviewer's comment. We have referenced more articles.

They also don't do an adequate (by that I mean mainly consistent) job in linking the microbiome metabolism of Trp to brain function. They describe what is possible without reviewing what may actually occur (or what has been shown to occur). They do refer to a few situations where alteration of microbiomes does affect KP metabolite levels, but surely there is much more they could present.

In conclusion, this is a good review article for what it provides, but there are deficiencies in areas where it doesn't provide adequate information. As the authors are aware, the area of microbiome physiology is rapidly developing, and good review articles along the way can direct research efforts. As an aside, I particularly enjoyed their Figures, mainly for their simplicity and effective communication of ideas as opposed to details.

Thanks to the reviewer for the comment.

Comments on the Quality of English Language

Overall the English is good, but there are several places where subject-verb inconsistencies are present. These can be resolved by careful editing.

Thank you for your comment, we have checked the linguistic correctness.

We greatly appreciate your precious time and all your assistance and support, and we would be delighted to receive any additional feedback that can contribute to the improvement of our work.

Round 2

Reviewer 1 Report

Comments and Suggestions for Authors

Dear Authors

Thanks for the revision!

Good Luck

Author Response

Dear reviewer,

we want to express our gratitude once more for your efforts in enhancing the manuscript's quality.

Best of luck!

Reviewer 3 Report

Comments and Suggestions for Authors

This is a revision of a review article I previously reviewed of the multiple interactions of metabolites produced in the kynurenine pathway (KP) that is one prominent pathway of bacteria in the microbiome. The authors have responded nicely to my prior criticisms and concerns. Specifically, they have added text and references and addressed questions related to blood-brain barrier permeability.

Their review continues to provide data illustrating the complexity of the KP metabolites and their potentials for modulating brain and body functions. It may turn out that the majority of KP metabolites modify mitochondrial functions through NAD+ availability, effects on oxidative stress and modulation of mitochondrial dynamics (fission, fusion). However, the answers to these possibilities await further research, and this review article should stimulate many investigations into mitochondrial functions and KP.

The sections on mitochondrial diseases remain a weakness, but the thrust of this paper is not on mitochondrial diseases, rather the KP pathway and microbiome contributions that may influence mitochondrial functions and thus mitochondrial diseases. Thus the authors can be "forgiven" somewhat on their approach to mitochondrial diseases, but perhaps a few references to more comprehensive reviews are in order.

Overall I found that this revision reads better than the original version, and I again compliment the authors on their Figures that are very educational. With a few references to more comprehensive reviews of mitochondrial diseases, I feel that this paper is now acceptable.

Comments on the Quality of English Language

I still note a few subject-verb discordancies that are easily corrected. "Crohn's" disease is misspelled twice on lines 237-238,

Author Response

Dear Reviewer,

Thank you once more for your contributions to enhancing the manuscript's quality. We have rectified the highlighted typo and incorporated new references into the section on mitochondrial diseases.

All the best!